# *C. elegans* is not a robust model organism for the magnetic sense

Erich Pascal Malkemper [1✉], Patrycja Pikulik[1,2,4], Tim Luca Krause[1,4], Jun Liu[3], Li Zhang [1], Brittany Hamauei[1] & Monika Scholz [3✉]

Magnetoreception is defined as the ability to sense and use the Earth's magnetic field, for example to orient and direct movements. The receptors and sensory mechanisms underlying behavioral responses to magnetic fields remain unclear. A previous study described magnetoreception in the nematode *Caenorhabditis elegans*, which requires the activity of a single pair of sensory neurons. These results suggest *C. elegans* as a tractable model organism for facilitating the search for magnetoreceptors and signaling pathways. The finding is controversial, however, as an attempt to replicate the experiment in a different laboratory was unsuccessful. We here independently test the magnetic sense of *C. elegans*, closely replicating the assays developed in the original publication. We find that *C. elegans* show no directional preference in magnetic fields of both natural and higher intensity, suggesting that magnetotactic behavior in the worm is not robustly evoked in a laboratory setting. Given the lack of a robust magnetic response under controlled conditions, we conclude that *C. elegans* is not a suitable model organism to study the mechanism of the magnetic sense.

[1] Max Planck Research Group Neurobiology of Magnetoreception, Max Planck Institute for Neurobiology of Behavior – caesar, Bonn, Germany. [2] Department of Game Management and Wildlife Biology, Faculty of Forestry and Wood Sciences, Czech University of Life Sciences, 16521 Prague 6, Czech Republic. [3] Max Planck Research Group Neural Information Flow, Max Planck Institute for Neurobiology of Behavior – caesar, Bonn, Germany. [4]These authors contributed equally: Patrycja Pikulik, Tim Luca Krause. ✉email: pascal.malkemper@mpinb.mpg.de; monika.scholz@mpinb.mpg.de

Magnetoreception is the ability of organisms to sense the weak Earth's magnetic field. Behavioral evidence for magnetoreception is available for many species, but the primary sensory cells remain to be discovered[1]. A central problem hampering progress on the neuronal level is the lack of a genetically accessible model species with a robust magnetic response.

A seminal study by Vidal-Gadea et al.[2] promised a revolutionary addition to the palette of magnetotaxis model organisms. The authors reported directional magnetic responses of the widely used nematode *Caenorhabditis elegans* to magnetic fields. The evidence was based on three behavioral assays: (1) The worms moved up- or downwards in a vertical agar-filled pipette, (2) at a certain angle to the magnetic field on a horizontal agar plate, and (3) towards a strong magnetic anomaly created by a neodymium magnet placed underneath a horizontal agar plate. The latter assay promised to be particularly powerful as it allowed the fast screening of mutants for magnetic sensitivity. Employing this assay, Vidal-Gadea et al. (2015) screened sensory mutants to reveal a critical role of a pair of AFD sensory neurons in magnetosensation. Finally, using calcium imaging, the authors reported that AFD neurons responded to magnetic stimuli.

The identification of a pair of primary sensory neurons is promised to enable the systematic investigation of the subcellular sensory mechanism for magnetoreception. Motivated by this promise, others set out to replicate the behavioral magnetic orientation assays in *C. elegans*, but with varied success. Independent studies by Njus et al.[3] and Landler et al.[4,5] failed to identify magnetic responses under carefully controlled magnetic conditions, while further studies from the original authors in collaboration with other labs reported successful replications[6,7]. An independent positive replication study that does not include authors from the original study and demonstrates magnetic orientation in *C. elegans* has, however, not been published. Intrigued by the simplicity of the assays developed by Vidal-Gadea et al. (2015), we attempted to replicate two of their magnetic orientation assays. We paid attention to the factors that influence magnetic orientation in *C. elegans*[7,8] and adhered to standards in the field of magnetic orientation including strict blinding of the experimenters and the use of double-wrapped magnetic coils[9–11].

We find that under these conditions the worms moved randomly on horizontal plates placed either on top of a strong neodymium magnet or within a homogenous Earth-strength horizontal magnetic field. In positive chemotaxis control experiments, however, we observed a strong directional preference. We conclude that even if *C. elegans* should have a magnetic sense, it is neither a suitable nor tractable genetic model organism to search for the magnetoreceptors.

## Results

**No magnetotaxis in millitesla magnetic fields.** We first set out to replicate the high-throughput plate assay described by Vidal-Gadea et al. (2015), which tests for directional preferences of horizontally migrating worms on an agar-filled plate with a local magnetic anomaly a thousand times stronger than the Earth's magnetic field. We allowed well-fed animals to migrate outwards from the center of an agar plate placed on top of a strong neodymium magnet and a non-magnetic metal (aluminum) disc on the opposite side (Fig. 1a). The magnetic field intensity on the agar surface directly above the magnetic disc was ~50 mT (500 Gauss), approximately 1000 times stronger than the local Earth-strength field of 49 µT. After 30 min, animals that had moved to either side were counted and a preference index was calculated for each plate. For the N2 strain, the preference index for plates with a magnet did not differ from non-magnetic control plates (Fig. 1b, Mann–Whitney $U$ test; $n_{Magnet} = 23$, $n_{Control} = 22$ plates,

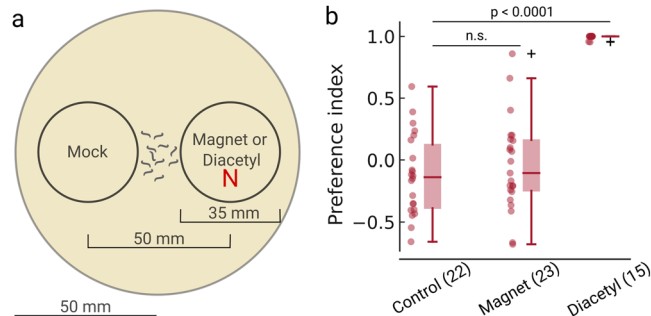

**Fig. 1 Strong neodymium magnets do not elicit magnetotaxis in *C. elegans* N2 animals. a** Schematic of the assay plate. Each circle contained the paralytic sodium azide to immobilize animals that reached one of the targets. **b** Preference index of the animals after exposure to either a non-magnetic aluminum disc (Control), a neodymium magnet (Magnet), or a known chemical attractant (Diacetyl). The boxplots follow Tukey's rule where the middle line indicates the median, the box denotes the first and third quartiles, and the whiskers show the 1.5 interquartile range above and below the box. Significance was assessed using a two-tailed Mann–Whitney $U$-test. Numbers in parentheses denote the number of independent replicates ($n$ = biologically independent experimental plates).

$U = 225.0$, $p = 0.532$). We repeated the experiments with an ancestral strain of N2, a strain that had been cryopreserved earlier than the standard N2 isolate (~10 generations from the original N2 isolated by S. Brenner) and should behave closer to wild isolates. We also found no preference or aversion of the neodymium magnet for the ancestral strain (Supplementary Fig. 1, Mann–Whitney $U$ test; $n_{Magnet} = 7$, $n_{Control} = 14$ plates, $U = 60.0$, $p = 0.433$). Measurements of the radio frequencies in the incubator in which the worms were grown and in the experimental room revealed low intensities that would not be expected to interfere with the magnetic sense[10] (Supplementary Fig. 2).

We also performed a positive control experiment where one of the circles contained the chemical attractant diacetyl, while the opposite side contained distilled water. As previously described[11], both the N2 strain (Mann–Whitney $U$ test; $n = 15$ plates, $U = 0.0$, $p < 0.0001$) and the ancestral N2 strain (Mann–Whitney $U$ test; $n = 7$ plates, $U = 0.0$, $p = 0.002$) displayed strong chemotaxis towards the odorant (Fig. 1b and Supplementary Fig. 1). We thus successfully replicated a published chemotaxis experiment in our laboratory.

**No directional preference in Earth-strength horizontal magnetic fields.** We next set up the horizontal field assay described by Vidal-Gadea et al.[2] to test whether *C. elegans* shows directional responses under natural field strengths. Agar plates with starved N2 worms at the center were placed in a horizontal magnetic field or a zero magnetic field. We used 65 µT total field intensity, in which starved N2 worms are expected to migrate at an angle of ~305° to the field lines[2]. We performed the assay in a triple-wrapped magnetic coil system that was placed on a vibration-decoupled platform in a Faraday cage (Fig. 2a). Electromagnetic fields in the radiofrequency range were of very low intensity, indicating a clean magnetic environment within the Faraday cage and the incubator in which the worms were grown (Supplementary Figs. 2 and 3).

We scored the horizontal migration of 12,019 worms after they were paralyzed at the margin of the plate (Fig. 2b). We excluded plates with low movement activity where less than 30 worms were scored and confined our analysis to plates that were tested under conditions that were discussed to promote magnetic orientation in *C. elegans*[7]. This included plates tested at a humidity below

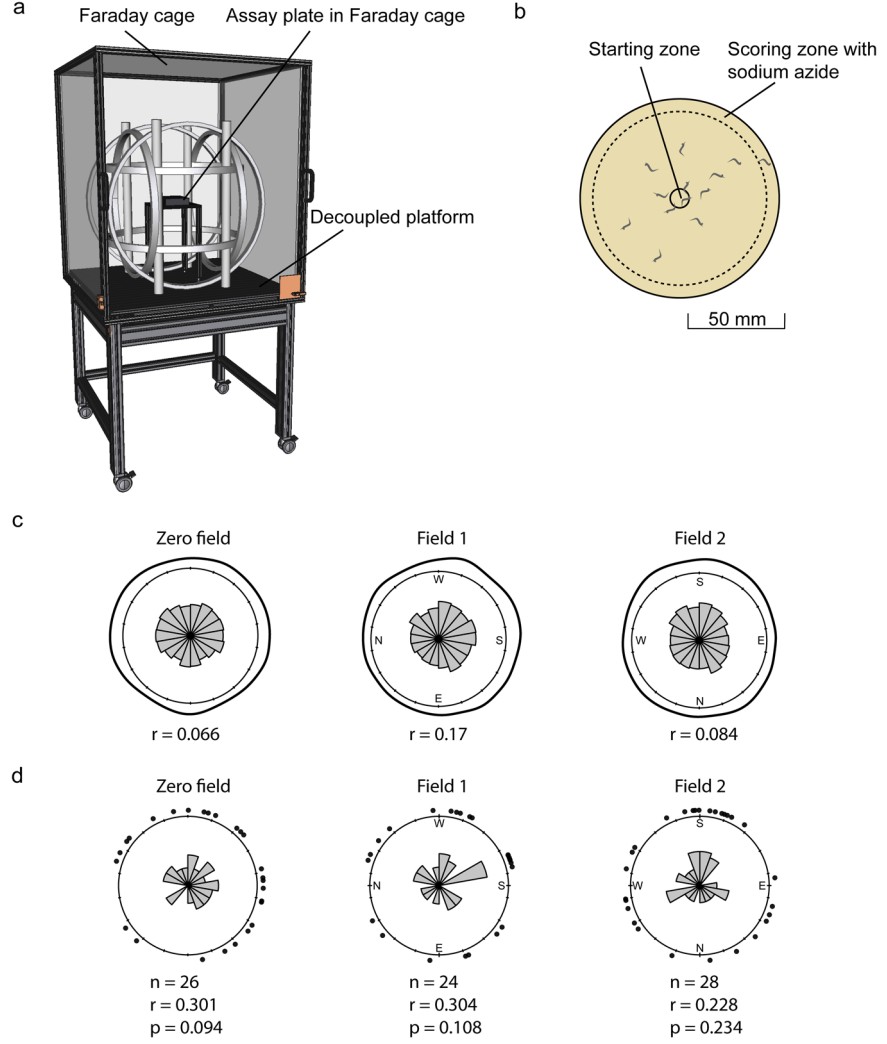

**Fig. 2 No directional preference in worms migrating in a horizontal Earth-strength magnetic field. a** Overview of the setup showing the Faraday cage with the Helmholtz coil system on a decoupled platform to minimize the transduction of vibrations. Assay plates were placed inside a grounded aluminum box in the center. **b** Schematic of the assay plate. Worms were released at the center and scored after being paralyzed at the periphery. **c** Distribution of the headings of individual worms in the three magnetic conditions. **d** Statistical analysis after calculating means for each plate. Each dot refers to the mean direction for one plate, the histogram in the center shows the observed frequency in 20-degree bins. Significance was assessed using the Rayleigh test. Sample size $n$ = biologically independent experimental plates. We observed no significant directional preference in any of the three magnetic conditions. Plots are shown in the same topographic orientation, the cardinal points indicate the direction of the magnetic field lines.

50%, a temperature of maximal 25 °C, a temperature difference between the start and end of the assay below 2 °C and an assay length of 55–65 min. We verified that filtering for these conditions did not unbalance the sample size between the groups.

Rose plots and density kernels of the individual headings did not indicate any directional preference in the two magnetic conditions (Fig. 2c). Worms migrating on a plate can influence each other and, therefore, cannot be treated as independent statistical units[4]. Using plates as individual units did not reveal a significant deviation from a random distribution in any of the three magnetic conditions (Fig. 2d, Rayleigh test, Zero field: $n = 26$, $r = 0.301$, $Z = 2.36$, $p = 0.094$; Field 1: $n = 24$, $r = 0.304$, $Z = 2.22$, $p = 0.108$, Field 2: $n = 28$, $r = 0.228$, $Z = 1.46$, $p = 0.234$). Furthermore, the $r$ (directedness) value of the zero field condition ($r = 0.301$) was included in the bootstrapped confidence intervals of both magnetic conditions (Field 1: 0.128–0.547, Field 2: 0.086–0.467), indicating that there was no significant increase in orientation strength in magnetic fields. In summary, we did not find evidence for orientation with respect to magnetic field lines in *C. elegans*.

## Discussion

We report that in our laboratories the nematode *C. elegans* does not display magnetotaxis. Despite carefully controlling the environment (temperature, humidity, magnetic and electric fields) and the state of the animals (age, developmental stage, feeding state), we were unable to detect a directional preference. The laboratories with previous reports of positive experiments were in Austin Texas, USA, and Buenos Aires, Argentina, respectively. It might be that another unnoticed environmental factor played a role, that goes beyond humidity, magnetic field, and temperature which we controlled for. Also, slight variations in the equipment and chemicals used might have led to the negative outcome, as batches and manufacturers will differ between labs on different continents. Reports of magnetoreception in other species show that effects are often small and likely this sense is secondary when other, often more prominent cues are available[12]. Thus animals who rely more on the magnetic sense might be a more suitable choice for these experiments.

It is interesting to speculate what purpose a magnetic sense might serve an animal that predominantly lives in compost, navigates over

centimeters, and whose dispersion strategy is to attach itself to slugs, flying beetles or other insects[13–15]. Vidal-Gadea et al.[2] suggested the inclination of the Earth's magnetic field helps the animals to guide vertical movement towards the surface or into the ground, depending on their feeding state. A similar kind of magnetotaxis is displayed by magnetotactic bacteria that are passively aligned with the Earth's magnetic field lines, which helps them to stay in a microaerobic zone just above the sediment. The taxis along the magnetic field lines is beneficial because they cannot make use of gravity. This is different for *C. elegans*, however, which has been shown to exhibit positive gravitaxis, indicative of a sense of gravity[16,17]. Landler et al.[4] highlighted further conceptual issues with the hypothesis by Vidal-Gadea et al.[2]. Finally, *C. elegans* navigate using light- and oxygen levels, and these cues elicit strong, well-established locomotion responses[18,19]. Similarly, our chemotaxis assay showed large effect sizes and resulted in clear preferences even for smaller sample sizes. If the magnetic field is used for navigation, one would expect similarly strong locomotion responses when it is the only directional cue available.

Robust behavioral assays in combination with genetics is the strength of the *C. elegans* model system and they have been key to understanding the mechanisms underlying odor- and chemotaxis. While our data does not disprove that *C. elegans* possess a magnetic sense, magnetic orientation is not robustly evoked in a laboratory setting. In line with reports of absent magnetic responses by other labs[3,4], we, therefore, conclude that *C. elegans* is unfortunately not a tractable, genetic model organism to study the mechanism of the magnetic sense.

## Methods

**Animals**. *Caenorhabditis elegans* wild-type N2 (RRID:WB-STRAIN:WB-Strain00000001) and N2 ancestral (RRID:WB-STRAIN:WBStrain00000003) strains were provided by the Caenorhabditis Genetic Center (University of Minnesota, USA). Worms were maintained on NGM agar plates seeded with *Escherichia coli* OP50 strain as a food source. Animals were cultured at always the same position inside an incubator (KB400; Binder, Tuttlingen, Germany) in the dark at a stable temperature of 20 °C in an Earth-strength magnetic field (40–75 µT). All worms used in assays were synchronized by bleaching healthily growing gravid adults and growing eggs in M9 buffer overnight on a nutating mixer until larval arrest at L1 to obtain an age-synchronized population. L1 were transferred to 6 cm plates seeded with OP50 *E. coli* and allowed to develop for 66 h in the incubator until they reached the young adult stage. All behavioral assays with worms specified as 'fed' were never starved and started within 10 min of their wash-off from seeded plates. Tests with 'starved' nematodes were performed after keeping them in M9 buffer for at least 30 min.

**Replication protocols**. The protocols used were copied as closely as possible from the original study, using the same media, same genotype, same developmental state, same range of environmental conditions as specified in ref. [7]. To maintain the stable culturing temperature as prescribed and to guarantee the same developmental timing we grew animals in an incubator in a plastic box in a specific, constant location. We report the static and time-varying electromagnetic fields at this location. The detailed behavioral testing protocols used below also follow the directions given in ref. [2] with the environmental factors as detailed further in ref. [7]. We made the following minor change to the protocol: Animals were transferred from culturing plates to testing plates using washes instead of picking with a metal pick. This allowed us to keep a very tight timing schedule with large numbers of animals as required by the original study.

**Animal preparation**. Culture plates containing bleaching-synchronized, on-food day-1 adults were washed off with 1 ml of M9 buffer. The nematode's suspension was poured into a 1.5 ml test tube and left to sit at room temperature (approx. 21 °C) for around 2 min to allow the worms sink to the bottom. The supernatant was removed and 1 ml of M9 was added to the precipitate to rinse off *E. coli* residue. This wash was repeated 3 times. After the third wash excess liquid was removed and 10 µl of the sample (50–100 hermaphrodites) were pipetted onto the center of the taxis plates.

**Taxis plates**. Taxis plates were prepared from 10 cm NGM plates (1-day old, 1.7% agar) by marking the center ('start') and two 3.5 cm diameter circles indicating the 'finish' points on opposite sides of the plate. The centers of the circles were 5 cm apart. In the center of the circles 1 µl of 1 M sodium azide (Sigma-Aldrich #S2002) was added to paralyze animals as soon as they reached one circle.

**Magnetotaxis assays**. To avoid any fluctuation in environmental conditions, control and test plates were always run simultaneously. The test plates were placed on top of a 3.5 cm diameter neodymium magnet (3.5 cm diameter, www.supermagnete.de S-35-05-N) with the magnetic north facing up on one side and a non-magnetic aluminum disc of similar appearance and dimensions on the other side. The intensity of the disc magnet was measured with a hall effect gaussmeter (FW Bell 5180 with SAD18-1904 axial probe). Control plates (C - Control) were placed on two identical aluminum disks. The assay was enclosed in opaque styrofoam boxes on separate benches to light or temperature gradients. All assays were run in darkness. The location for each condition was randomized to minimize local influences by factors other than the magnetic field. To start the assay, worms were released from the drops by draining the buffer with filter paper. The experiment was started within 10 min of depriving the worms of their food source, so that the animals were fed and would not change their behavior due to starvation[7]. After 30 min, most of the worms were trapped in one of the two end areas. An experimenter blinded to the experimental condition counted the number of paralyzed animals in each circle using a dissection microscope. After unblinding, the Preference Index (PI) was calculated as

$$\text{PI} = \frac{T - C}{T + C} \quad (1)$$

where $T$ is the number of worms in the area with the neodymium magnet, and $C$ is the number of worms in the area with the non-magnetic metal control (Vidal-Gadea et al. 2015).

**Chemotaxis assays**. Chemotaxis plates were prepared similarly to the magnetotaxis plates, with a strong odorant replacing the neodymium magnets. To attract the worms, 1 µl of 0.5% aqueous diacetyl solution (Acros Organics #107650050; T - taxis) was added to one circle and 1 µl of $ddH_2O$ to the other circle (C - control). The excess M9 buffer was carefully removed with a small piece of filter paper to allow the worms to crawl on the agar surface. The plates were placed in styrofoam boxes to ensure a constant temperature and darkness and left for 30 min until counting. After allowing the animals to migrate for 30 min, the paralyzed worms were counted manually by a blinded experimenter as in the magnetotaxis assays.

**Horizontal field assay**. Sodium azide (0.1 M, 15 µl) was applied on the agar surface along the edges of unseeded NGM agar plates (1-day old, 10 cm diameter, 1.7% agar) using a pipet tip. After being washed off the culturing plates (according to the protocol described in 'Magnetotaxis assay') synchronized hermaphrodites were left in 1 ml of M9 buffer for 30 min to starve. After the starvation period, the supernatant was removed and a 10 µl aliquot of 50–100 worms was released in the center of the assay plate. Once the liquid was carefully removed with filter paper, the sample was placed inside the Faraday cage located at the center of the magnetic coil system.

The behavioral assays were conducted in total darkness under three magnetic conditions: a horizontal Earth-strength magnetic field (65 µT total intensity), the same field rotated by 90°, and a near zero magnetic field (<100 nT total intensity). The worms were allowed to migrate for one hour, after which the plates were removed from the coil system and all immobilized animals at the rim of the plate were marked under a dissection microscope. The plates were photographed with a compass background and the angles were measured using an image viewing program. The experimenter marking the plates and analyzing the angles was blinded to the magnetic condition of the experiment. To tightly control the environmental conditions, temperature, and humidity were recorded at the start and end of each experiment. Data was collected from February 2021 to March 2022.

**Statistics and reproducibility**. Preference indices were statistically analyzed in Python using a non-parametric two-sided Mann–Whitney U-test which does not require a normal distribution of the data. No adjustment for multiple-testing was applied.

Circular statistics were performed in R using the *circular* package. For raw data plots, we computed the kernel density estimates using the density.circular function with a smoothing bandwidth of 40. Worms on a plate can interact and influence each other and, therefore, represent non-independent samples[4]. We thus calculated the mean vector length $r$ and direction $\mu$ for each plate by vector summation of the individual headings. We then calculated second order means by vector summation of the plate means and tested for statistical significance using the Rayleigh test[20]. Only plates where at least 30 worms reached the periphery of the plate were included in the analysis.

To test for a difference in the directedness between the near zero magnetic field group and the other two magnetic treatments, we employed a bootstrap technique[21]. For this, we tested if the length of the mean resultant vector ($r$-value) of the zero group falls within the confidence intervals of the $r$-values of the magnetic groups. A random subsample of n orientation angles was drawn with replacement from the experimental groups ($n = 26$ for zero, $n = 24$ for field 1, $n = 28$ for field 2). Then the corresponding $r$-value was calculated based on these orientation angles. This procedure was repeated with a new randomization 100,000 times. The resulting 100,000 $r$-values were ranked lowest to highest after which the $r$-values at the ranks 2500 and 97,500 define the 95% confidence limits for the

observed $r$-value of the tested group. If the $r$-value observed in the zero magnetic field group lay outside these confidence intervals, the magnetic group was significantly more directed than the zero group with a significance of $p < 0.05$.

**Magnetic coil set up**. To generate or reset a horizontal magnetic field we used a tripled-wrapped three axial Helmholtz Coil system (Claricent, Munich), which was located in the center of a custom-built grounded aluminum Faraday cage with the dimensions 84 cm (length) × 86 cm (width) × 99 cm (height). One of the three coil wraps was used to cancel the ambient magnetic field while the other two wraps either produced (with parallel current flow) or did not produce (antiparallel, near zero condition) a horizontal magnetic field. This ensured that the current flow, and thus also possible side effects such as heat production and vibrations were identical in the three magnetic conditions. Such a controlled magnetic environment cannot be created with double-wrapped coils if no static magnetic field shielding (e.g., a mu-metal chamber as reported by ref. [5]) is available. The magnetic coil diameters were 61 cm ($z$-axis), 63 cm ($x$-axis), and 69 cm ($y$-axis). The coils were electrically shielded to minimize the emission of electric fields and they were powered by two low-ripple programmable power supplies (Rohde & Schwarz Hameg HMP4040 and HMP4030). Static magnetic fields of Earth-strength were measured using a high-sensitivity three-axis fluxgate magnetometer (Meda FVM400). The coils and behavioral setup were placed on a custom-made anti-vibration platform that served as the floor of the Faraday cage. The test plates were placed on top of a non-magnetic platform inside a grounded aluminum chamber which served as an additional Faraday cage that blocked ambient light.

**Measurements of time-dependent electromagnetic fields**. The magnetic and electric components of time-dependent (radiofrequency) electromagnetic fields were measured individually with two different antennas connected to a spectrum analyzer (Tektronix RSA306B Real-Time Spectrum Analyzer 9 kHz–6.2 GHz). For each condition we measured the fields for a period of 20 min, the signal analyzer was set to 'max hold' and a resolution bandwidth of 10 kHz (to allow comparisons with the measurements presented by Engels et al. 2014). The traces are based on 64,001 measurement points between 9 kHz and 30 MHz. Measurements were done at times of day similar to the worm experiments, but not during the testing runs to avoid influencing the experiments. The magnetic component between 9 kHz and 30 MHz was measured with a calibrated active loop antenna (Schwarzbeck Mess-Elektronik, FMZB 1513, 9 kHz–30 MHz frequency response). The electric component between 9 kHz and 30 MHz was measured with a calibrated active biconical antenna (Schwarzbeck Mess-Elektronik, EFS 9218, 9 kHz–300 MHz frequency response).

**Reporting summary**. Further information on research design is available in the Nature Portfolio Reporting Summary linked to this article.

## Data availability

The data that support the findings are available here: https://osf.io/h2qm5/.

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

## Acknowledgements

E.P.M. receives funding from the European Research Council (ERC) under the European Union's Horizon 2020 research and innovation program (grant agreement No 948728). The project iBEHAVE (M.S. and E.P.M.) has received funding from the program "Netzwerke 2021", an initiative of the Ministry of Culture and Science of the State of Northrhine Westphalia. The sole responsibility for the content of this publication lies with the authors. P.P. was supported by the European ERASMUS + traineeship program. We are thankful to Rolf Honnef (MPINB mechanical workshop) for constructing the Faraday cage and providing the technical drawing. Some strains were provided by the CGC, which is funded by NIH Office of Research Infrastructure Programs (P40 OD010440).

## Author contributions

Conceptualization: M.S. and E.P.M.; Formal analysis: M.S. and E.P.M.; Funding acquisition: M.S. and E.P.M.; Investigation: P.P., T.L.M., L.Z., J.L., and B.H.; Resources: J.L., M.S., and E.P.M.; Supervision: E.P.M. and M.S.; Visualization: E.P.M. and M.S.; Writing – original draft: E.P.M. and M.S.; Writing - review & editing: P.P., T.L.K., J.L., L.Z., B.H., E.P.M., and M.S.

## Funding

## Competing interests

The authors declare no competing interests.
