## [Peer Review File · Communications Biology]

Reviewers' comments:

Reviewer #1 (Remarks to the Author):

In the manuscript "C. elegans is not a robust model organism for the magnetic sense", the authors set out to replicate the previously claimed magnetoreceptive capabilities of C. elegans. This species is particularly interesting for the magnetoreception community as, in contrast to many other species, in which magnetoreceptive capabilities have been shown, it provides easy access to reasonable numbers of animals, but most importantly, it represents a potential genetic model system. This could be of fundamental importance in the magnetoreception field, since the underlying mechanisms in any organism remain elusive to date.

In their study, the authors used the "Vidal-Gadea"-approach, which appeared as an intriguingly robust, but simple, setup to test magnetic sensitivity in C. elegans. In their experiments, the worms did not show magnetic field-invoked directed migratory responses under both natural and approx. 1000 times Earth's field intensities, suggesting that magnetotactic behavior in the worm is not robustly evoked, at least in their laboratory setting. Instead, the authors report strong chemotaxis towards the odorant diacetyl, which generally validates the functionality of their setup.

I assume, the authors would have preferred to see a "positive" result in a promising new model species, but that's science, isn't it? The field of magnetoreception has faced various throwbacks in the last decade, which seriously question the existence of a previously claimed magnetic sense, and this one seems to be the next one on this list. Nevertheless, the last decade has also shown the importance to publish negative data, as it forces the field to rethink what has previously claimed to be true.

I see no methodological flaws in this study. The authors

- 1) performed strictly double blinded protocols
- 2) they used the by now gold standard in magnetoreception research, i.e. double-wound magnetic coils
- 3) statistical analysis is sound, and, regarding any potential magnetic field effects, far from showing the slightest bit of a directional tendency
- 4) the number of tested animals is exceptional (12.000!!!)
- 5) singly tested worms did also not show any magnetic field-dependent directional preference ruling out the possibility that worms could have influenced each other's migratory behaviour.
- 5) any potential artifacts in the magnetic conditions have been measured and ruled out in the supplemental data.

Admittedly, the attempt to replicate the Vidal-Gadea approach is not new. As the authors correctly report, other labs have also tried to replicate the Vidal-Gadea experiments without success. Given the lack of a robust magnetic response under unprecedentedly controlled conditions, it can be concluded that C. elegans is not a suitable model organism to study the mechanism of the magnetic sense.

I would be pleased to see this paper being published in Communications Biology as it is.

Reviewer #2 (Remarks to the Author):

Review on the manuscript: C. elegans is not a robust model organism for the magnetic sense
E. Pascal Malkemper, Patrycja Pikulik, Tim L. Krause, Jun Liu, Li Zhang, Brittany Hamauei and Monika Scholz

For the Communications Biology journal
Brief summary of the manuscript

I completely agree with the authors: A central problem hampering progress on the neuronal level is the lack of a genetically accessible model species with a robust magnetic response. Authors set out to independently test the magnetic sense of *C. elegans*, closely replicating the assays developed in the publication (Vidal-Gadea et al., 2015 and Bainbridge et al., 2019) that reported "robust and tractable system to study magnetoreception". Malkemper et al. found that the worms moved randomly, and they conclude that even if *C. elegans* should have a magnetic sense, it is neither a suitable nor tractable genetic model organism to search for the magnetoreceptors.

Overall impression of the work

In the field of magnetoreception, independent replications are very much needed. Especially if a powerful genetic model was used in the original work and a robust phenotype was reported. Compared to the original papers, Malkemper et al. added some measurements that were not included in the original papers, such as analysis of electromagnetic background noise and positive control using the chemotaxis assay. They introduced blinded evaluation and corresponding modern techniques. The value of the present work and the resulting manuscript is high, and I believe that it should be published after minor revisions to the text.

Specific comments

I believe that in replication or verification such as this, it should be explicitly stated whether an identical protocol as close to the original as possible was used. If any methodological changes were made, please describe specifically what they were and why. Since the paper questions the "robustness" of the original work, not the magnetic susceptibility of *C. elegans* itself, it does not need to be extremely specific or detailed. However, given that this is the third replication attempt in a row, readers may wish to be explicitly informed about what the authors believe is the reason why the data is controversial. I do not at all recommend downplaying the original results without giving reasons, but just still providing some arguments trying to explain the controversy.

Minor comments:

- Page 1: "Magnetoreception is the ability to sense and use the Earth's magnetic field to orient and direct movement." Exactly what animals use their ability to perceive the MGF for, we do not yet fully know. Sometimes they don't move at all but still have magnetically oriented bodies, and other times their circadian clocks are affected by the MGF. Therefore, I only recommend that the definition of magnetoreception should not be understood in this somewhat narrow way.
- Page 2: Studies by Njus et al. (2015) and Landler et al. (2018a) are ranked as independent but it is stated that: "A truly independent replication study demonstrating magnetic orientation in *C. elegans* has not been published." Could it be more clearly stated what "truly" means? Please, explain why and how much your work differs from what was already published.
- Authors surely have considered the paper: Vidal-Gadea, A., Bainbridge, C., Clites, B., Palacios, B. E., Bakhtiari, L., Gordon, V., & Pierce-Shimomura, J. (2018). Response to comment on "Magnetosensitive neurons mediate geomagnetic orientation in *Caenorhabditis elegans*". *eLife* 7, e31414. doi: 10.7554/eLife.31414. It should be mentioned and cited here as well.

Response to Reviewers

We are very pleased that the referees view our paper positively, appreciating the value of using our work as guidance for future endeavors. We addressed all comments and questions raised by the referees below and marked our changes in the manuscript. In addition to responding to the reviewers comments we have implemented minor changes to match the journal's requirements as specified in the reporting summary sheet.

Reviewer #1

In the manuscript "C. elegans is not a robust model organism for the magnetic sense", the authors set out to replicate the previously claimed magnetoreceptive capabilities of C. elegans. This species is particularly interesting for the magnetoreception community as, in contrast to many other species, in which magnetoreceptive capabilities have been shown, it provides easy access to reasonable numbers of animals, but most importantly, it represents a potential genetic model system. This could be of fundamental importance in the magnetoreception field, since the underlying mechanisms in any organism remain elusive to date.

In their study, the authors used the "Vidal-Gadea"-approach, which appeared as an intriguingly robust, but simple, setup to test magnetic sensitivity in C. elegans. In their experiments, the worms did not show magnetic field-invoked directed migratory responses under both natural and approx. 1000 times Earth's field intensities, suggesting that magnetotactic behavior in the worm is not robustly evoked, at least in their laboratory setting.

Instead, the authors report strong chemotaxis towards the odorant diacetyl, which generally validates the functionality of their setup.

I assume, the authors would have preferred to see a "positive" result in a promising new model species, but that's science, isn't it? The field of magnetoreception has faced various throwbacks in the last decade, which seriously question the existence of a previously claimed magnetic sense, and this one seems to be the next one on this list. Nevertheless, the last decade has also shown the importance to publish negative data, as it forces the field to rethink what has previously claimed to be true.

I see no methodological flaws in this study. The authors

- 1) performed strictly double blinded protocols
- 2) they used the by now gold standard in magnetoreception research, i.e. double-wound magnetic coils
- 3) statistical analysis is sound, and, regarding any potential magnetic field effects, far from showing the slightest bit of a directional tendency
- 4) the number of tested animals is exceptional (12.000!!!)
- 5) singly tested worms did also not show any magnetic field-dependent directional preference ruling out the possibility that worms could have influenced each other's migratory behaviour.

5) any potential artifacts in the magnetic conditions have been measured and ruled out in the supplemental data.

Admittedly, the attempt to replicate the Vidal-Gadea approach is not new. As the authors correctly report, other labs have also tried to replicate the Vidal-Gadea experiments without success. Given the lack of a robust magnetic response under unprecedentedly controlled conditions, it can be concluded that *C. elegans* is not a suitable model organism to study the mechanism of the magnetic sense.

I would be pleased to see this paper being published in *Communications Biology* as it is.

Response: We thank the reviewer for taking the time to review the manuscript and the positive assessment of our work.

Reviewer #2

Brief summary of the manuscript

I completely agree with the authors: A central problem hampering progress on the neuronal level is the lack of a genetically accessible model species with a robust magnetic response. Authors set out to independently test the magnetic sense of *C. elegans*, closely replicating the assays developed in the publication (Vidal-Gadea et al., 2015 and Bainbridge et al., 2019) that reported “robust and tractable system to study magnetoreception”. Malkemper et al. found that the worms moved randomly, and they conclude that even if *C. elegans* should have a magnetic sense, it is neither a suitable nor tractable genetic model organism to search for the magnetoreceptors.

Overall impression of the work

In the field of magnetoreception, independent replications are very much needed. Especially if a powerful genetic model was used in the original work and a robust phenotype was reported. Compared to the original papers, Malkemper et al. added some measurements that were not included in the original papers, such as analysis of electromagnetic background noise and positive control using the chemotaxis assay. They introduced blinded evaluation and corresponding modern techniques. The value of the present work and the resulting manuscript is high, and I believe that it should be published after minor revisions to the text.

Specific comments

I believe that in replication or verification such as this, it should be explicitly stated whether an identical protocol as close to the original as possible was used. If any methodological changes were made, please describe specifically what they were and why.

Response: We agree with the reviewer and we have added text in the methods section stating which protocols we followed and which (minor) changes were made for technical reasons.

“Replication protocols

The protocols used were copied as closely as possible from the original study, using the same media, same genotype, same developmental state, same range of environmental conditions as specified in [7]. To maintain the stable culturing temperature as prescribed and to guarantee the same developmental timing we grew animals in an incubator in a plastic box in a specific, constant location. We report the static and time-varying electromagnetic fields at this location. The detailed behavioral testing protocols used below also follow the directions given in [2] with the environmental factors as detailed further in [7]. We made the following minor change to the protocol: Animals were transferred from culturing plates to testing plates using washes instead of picking with a metal pick. This allowed us to keep a very tight timing schedule with large numbers of animals as required by the original study.”

Since the paper questions the "robustness" of the original work, not the magnetic susceptibility of *C. elegans* itself, it does not need to be extremely specific or detailed. However, given that this is the third replication attempt in a row, readers may wish to be explicitly informed about what the authors believe is the reason why the data is controversial. I do not at all recommend downplaying the original results without giving reasons, but just still providing some arguments trying to explain the controversy.

Response: We appreciate this suggestion and added the following sentences to the discussion:

“The laboratories with previous reports of positive experiments were in Austin Texas, USA and Buenos Aires, Argentina, respectively. It might be that another unnoticed environmental factor played a role, that goes beyond humidity, magnetic field and temperature which we controlled for. Also slight variations in the equipment and chemicals used might have led to the negative outcome, as batches and manufacturers will differ between labs on different continents. Reports of magnetoreception in other species show that effects are often small and likely this sense is secondary when other, often more prominent cues are available [11]. Thus animals who rely more on the magnetic sense might be a more suitable choice for these experiments.”

11. Johnsen S, Lohmann KJ, Warrant EJ. Animal navigation: a noisy magnetic sense? *J Exp Biol.* 2020;223. doi:10.1242/jeb.164921

Page 1: "Magnetoreception is the ability to sense and use the Earth's magnetic field to orient and direct movement." Exactly what animals use their ability to perceive the MGF for, we do not yet fully know. Sometimes they don't move at all but still have magnetically oriented bodies, and other times their circadian clocks are affected by the MGF. Therefore, I only recommend that the definition of magnetoreception should not be understood in this somewhat narrow way.

Response: We thank the reviewer for this reminder. Of course, magnetic orientation is not the only behavioral expression of a magnetic sense as the mentioned magnetic alignment and clock effects clearly demonstrate. We changed the sentence to:

“Magnetoreception is defined as the ability to sense and use the Earth's magnetic field, for example to orient and direct movements.”

Page 2: Studies by Njus et al. (2015) and Landler et al. (2018a) are ranked as independent but it is stated that: “A truly independent replication study demonstrating magnetic orientation in *C. elegans* has not been published.” Could it be more clearly stated what “truly” means? Please, explain why and how much your work differs from what was already published.

Response: We acknowledge that the wording was not clear here. We removed “truly” and instead explained what we meant by it, i.e. a positive replication which does not include any of the original authors. The new sentence reads:

“An independent positive replication study that does not include authors from the original study and demonstrates magnetic orientation in *C. elegans* has not been published.”

• Authors surely have considered the paper: Vidal-Gadea, A., Bainbridge, C., Clites, B., Palacios, B. E., Bakhtiari, L., Gordon, V., & Pierce-Shimomura, J. (2018). Response to comment on "Magnetosensitive neurons mediate geomagnetic orientation in *Caenorhabditis elegans*". *eLife* 7, e31414. doi: 10.7554/eLife.31414. It should be mentioned and cited here as well.

Response: We thank the reviewer for pointing out this omission. We are of course aware of this work and now add the reference here:

“We paid attention to the factors that influence magnetic orientation in C. elegans [6,7] and adhered to standards in the field of magnetic orientation including strict blinding of the experimenters and the use of double-wrapped magnetic coils [8].”

6. Bainbridge C, Clites BL, Caldart CS, Palacios B, Rollins K, Golombek DA, et al. Factors that influence magnetic orientation in *Caenorhabditis elegans*. *J Comp Physiol A Neuroethol Sens Neural Behav Physiol*. 2020;206: 343–352. doi:10.1007/s00359-019-01364-y
7. Vidal-Gadea A, Bainbridge C, Clites B, Palacios BE, Bakhtiari L, Gordon V, et al. Response to comment on “Magnetosensitive neurons mediate geomagnetic orientation in *Caenorhabditis elegans*.” *eLife*. 2018. doi:10.7554/eLife.31414
8. Kirschvink JL. Uniform magnetic fields and double-wrapped coil systems: improved techniques for the design of bioelectromagnetic experiments. *Bioelectromagnetics*. 1992;13: 401–411. doi:10.1002/bem.2250130507